# Clinical Application of Unidirectional Porous Hydroxyapatite to Bone Tumor Surgery and Other Orthopedic Surgery

**DOI:** 10.3390/biomimetics9050294

**Published:** 2024-05-15

**Authors:** Toshiyuki Kunisada, Eiji Nakata, Tomohiro Fujiwara, Toshiaki Hata, Kohei Sato, Haruyoshi Katayama, Ayana Kondo, Toshifumi Ozaki

**Affiliations:** 1Department of Medical Materials for Musculoskeletal Reconstruction, Okayama University Graduate School of Medicine, Dentistry, and Pharmaceutical Sciences, 2-5-1, Shikata-cho, Okayama 700-8558, Japan; 2Department of Orthopaedic Surgery, Okayama University Graduate School of Medicine, Dentistry, and Pharmaceutical Sciences, 2-5-1, Shikata-cho, Okayama 700-8558, Japan

**Keywords:** hydroxyapatite, bone tumor, orthopedic surgery, unidirectional porous hydroxyapatite, bone graft

## Abstract

Unidirectional porous hydroxyapatite (UDPHAp) was developed as a remarkable scaffold characterized by a distinct structure with unidirectional pores oriented in the horizontal direction and connected through interposes. We evaluated the radiographic changes, clinical outcomes, and complications following UDPHAp implantation for the treatment of bone tumors. Excellent bone formation within and around the implant was observed in all patients treated with intralesional resection and UDPHAp implantation for benign bone tumors. The absorption of UDPHAp and remodeling of the bone marrow space was observed in 45% of the patients at a mean of 17 months postoperatively and was significantly more common in younger patients. Preoperative cortical thinning was completely regenerated in 84% of patients at a mean of 10 months postoperatively. No complications related to the implanted UDPHAp were observed. In a pediatric patient with bone sarcoma, when the defect after fibular resection was filled with UDPHAp implants, radiography showed complete resorption of the implant and clear formation of cortex and marrow in the resected part of the fibula. The patient could walk well without crutches and participate in sports activities. UDPHAp is a useful bone graft substitute for the treatment of benign bone tumors, and the use of this material has a low complication rate. We also review and discuss the potential of UDPHAp as a bone graft substitute in the clinical setting of orthopedic surgery.

## 1. Introduction

Autogenous bone grafting is widely used in orthopedic surgery. Bone defects are sometimes too large to be filled with autogenous bone alone, and some complications associated with autogenous bone graft harvesting have been reported, such as donor site fracture and infection, prolonged operation time, and increased blood loss [1]. Various artificial bone graft substitutes with different compositions, porous structures, and porosities have been developed and used in orthopedic surgery [2]. Hydroxyapatite (HAp) has been widely used as a synthetic graft material for orthopedic surgery because of its sufficient strength, osteoconductive ability, and similarity to the mineral components of bone. However, the pores of implanted first-generation HAp are rarely filled with newly formed host bone, probably because of the closed structure of HAp with few interpore connections [3]. Therefore, new bone with HAp implants at defect sites may be fragile and prone to fracture. To overcome this disadvantage, second-generation porous HAp implants with adequate diameter interpore connections have been developed. Unidirectional porous hydroxyapatite (UDPHAp, REGENOS^®^, Kuraray Co., Ltd., Tokyo, Japan) has 75% porosity and 99.9% purity, with an interconnected porous structure [4]. Its most distinctive feature is that it consists of unidirectional oval pores oriented in the horizontal direction that completely penetrate the material (Figure 1). The unidirectional porous feature replicates the orientational structure of collagen and HAp of a long bone and increases the compressive strength of the bone in the longitudinal direction with a maximum of 13.1 MPa compared to the directionless porous structure [2]. The pore size (approximately 100–300 μm in the longest diameter) and microstructure can facilitate the invasion of cells and fluids necessary for osteogenesis. Because of these features, histologic analysis has been performed in animal studies to evaluate new bone formation, regeneration, and remodeling within the UDPHAp implants. Clinically, UDPHAp has been widely used to treat various bone defects in orthopedic surgery.

Benign bone tumors develop within the bone tissue and are typically slow growing. Some benign bone tumors can cause the thinning of the bone cortex, and the structural integrity of the affected bone is critical for pathological fracture. Benign bone tumors can be treated surgically, and it is important to fill the cavity after intralesional resection to restore mechanical strength and prevent pathological fracture. The defect after resection of a bone tumor may sometimes be too large to be filled with autogenous bone alone, and various materials have been used to fill these cavities [3,5]. In addition to bone tumor surgery, other orthopedic procedures such as fracture fixation, osteotomy, and laminoplasty often leave bone defects or gaps that should be treated with bone grafting. Early bone formation to restore bone defects or gaps is beneficial for patients to achieve better activities of daily living. In this article, we analyzed the radiographic and clinical outcomes of patients who underwent UDPHAp implantation after the surgical resection of bone tumors and reviewed papers showing the results of clinical cases treated with UDPHAp implantation in other orthopedic surgeries. We then discussed the clinical benefits of UDPHAp implantation in orthopedic surgery.

## 2. Clinical Application to Bone Tumor Surgery

We retrospectively analyzed 44 patients who underwent intralesional resection and UDPHAp implantation for benign bone tumors between 2010 and 2015 [6]. Patients who were surgically treated for local recurrence were excluded. There were 30 males and 14 females with a mean age of 24 years (range, 8–72 years). The most common histologies are simple bone cyst, enchondroma, giant cell tumor of bone, and fibrous dysplasia. Adequate amounts of both block- and granule-type UDPHAp were implanted into bone defects after tumor resection in 24 patients, and only granule-type UDPHAp was implanted in 20 patients. We evaluated radiographic changes, clinical outcomes, and complications after UDPHAp implantation for the treatment of benign bone tumors. Regular postoperative radiographs were taken in the outpatient clinic every 3–6 months. Two orthopedic surgeons who were blinded to the clinical conditions assessed radiographic changes in the implanted UDPHAp according to Tamai’s staging [3], which is divided into five stages based on bone formation in the implanted UDPHAp. Implanted UDPHAp showed diffuse extensive sclerosis in all patients, and the outlines of granular and block-type UDPHAp became indistinct on radiographs at an average of 12 months after surgery, indicating excellent new bone formation within and around the implanted UDPHAp (Figure 2). The absorption of UDPHAp and marrow space remodeling were observed in 45% of patients at a mean of 17 months postoperatively and were significantly more common in younger patients (Table 1). Early diffuse sclerosis due to new bone formation is a significant indicator of good implant resorption and bone remodeling. All nine preoperative pathological fractures healed completely within 3 months after surgery. Preoperative cortical thinning was completely regenerated in 84% of patients at an average of 10 months after surgery. Gender, age, location in the long bone, pathological fracture, and the volume of the UDPHAp implanted did not show significant associations with the regeneration of cortical thickness. Delayed wound healing, postoperative infection, and allergic reactions related to implanted UDPHAp were not observed. A patient with a simple bone cyst and a preoperative pathological fracture of the proximal femur fell down the stairs and developed a fracture 9 months after implant removal. The patient underwent another internal fixation and showed good bone healing at the final follow-up.

The results of this study have several important limitations. First, there was no comparison group or randomization. Second, three-dimensional imaging assessment was not performed. Computed tomography (CT) may be able to more accurately show new bone formation and the incorporation of implanted UDPHAp, but repeated CT assessment is not appropriate because of the ethical issue of radiation exposure. Third, because benign bone tumors generally occur in young patients, most patients were relatively young and may have better bone formation than older patients.

In a pediatric patient with Ewing’s sarcoma of the pelvis, a free fibular graft was harvested from the left leg for pelvic reconstruction after tumor resection [7]. After fibular resection, the defect was filled with column-shaped UDPHAp implants (Figure 3). The remaining periosteum was sutured to cover the implanted UDPHAp as completely as possible. A plain radiograph taken one month after surgery showed new bone formation in the gap between the remaining proximal fibula and the implanted UDPHAp and a callus-like structure around the centrally implanted UDPHAp. The resorption of the implanted UDPHAp was noted, and the partial remodeling of the marrow cavity was observed 11 months after surgery. There was good regeneration of the fibula with bone cortex and marrow 2.5 years after surgery. In addition, the bony continuity of the fibula in the segmental defect was complete after harvesting. The implanted UDPHAp was resorbed over time, and clear formation of cortex and marrow was observed in the resected part of the fibula at the final follow-up (12 years). The implantation of UDPHAp into the fibular defect did not cause any complications related to graft harvesting or UDPHAp implantation. The patient achieved good postoperative function and was able to walk without crutches and participate in sports activities. UDPHAp is a good bone substitute for filling segmental defects in the fibula after graft harvesting.

## 3. Clinical Application to Other Orthopedic Surgery

Porous HAp is widely used in orthopedic conditions due to its high osteoconductivity. High tibial osteotomy is a surgical procedure used to treat osteoarthritis or osteonecrosis of the knee. The gap left after open-wedge high tibial osteotomy can be filled with autologous, allogeneic, or artificial bone grafts. A retrospective study was performed to clinically and radiologically evaluate the availability, osteoconductivity, and resorption of UDPHAp used as an artificial substitute for open-wedge high tibial osteotomy in six patients [8]. The created gap was filled with UDPHAp and fixed with a plate. Block-type UDPHAp was cut to fit the gap, and granule-type UDPHAp was used to fill the tip of the gap. Radiographs and CT were evaluated up to 12 months postoperatively and showed affinity with the surrounding bone and increased sclerosis over time, suggesting good bone healing. Clinical evaluation using the Japanese Orthopaedic Association (JOA) knee score improved significantly after open-wedge high tibial osteotomy with UDPHAp. Osteogenesis progressed in and around the artificial bone grafts, indicating successful bone healing with UDPHAp. There was no gross displacement or collapse of the UDPHAp block graft within 12 months of surgery. The short-term results of the open-wedge high tibial osteotomy using UDPHAp as a bone graft substitute were satisfactory. UDPHAp was found to be safe and useful as a bone graft substitute for filling the gap in open-wedge high tibial osteotomy.

Open-door and double-door laminoplasties are surgical procedures commonly used to treat cervical myelopathy. HAp spacers are often used in laminoplasty to maintain the expanded position of the lamina. Thirty-nine patients underwent open-door laminoplasty with UDPHAp spacers, and radiographic and clinical outcomes were analyzed to elucidate the efficacy of UDPHAp spacers for open-door laminoplasty and the adverse events associated with their use [9]. Despite a good bone fusion rate of 87% on the hinge sides, postoperative CT assessment revealed the breakage and deformation of the implanted UDPHAp spacers in 69% of patients and laminar closure in 35% of patients. The clinical recovery rate of neurological symptoms according to the JOA score was low. In addition, there were some serious adverse events associated with their use. This study concluded that UDPHAp spacers are not suitable for open-door laminoplasty. In contrast, 50 patients underwent double-door laminoplasty using UDPHAp spacers, and the short-term bone bonding capacity of UDPHAp spacers used in double-door laminoplasty was evaluated [10]. Postoperative CT evaluation showed that the bone bonding rate was 67% at 12 months after surgery, and the change in the anterior–posterior diameter of the spinal canal was significantly greater for UDPHAp spacers than for autologous bone spacers. Although deformation of the implants was observed in 21% of patients, there was no evidence of breakage along their long axis on axial CT images. Clinical evaluation showed favorable neurological outcomes and functional improvements. In contrast to the conclusion reached in the analysis of UDPHAp spacers for open-door laminoplasty, the study concluded that UDPHAp spacers for double-door laminoplasty reduced the risk of early post-implantation dislocation and facilitated bone bonding through the infiltration of surrounding tissue, supporting the efficacy of UDPHAp spacers in double-door laminoplasty.

Bone defects sometimes remain after intra-articular fracture reduction, and bone grafting is often required to maintain the articular surface even with internal fixation. In a clinical case report, a patient who underwent surgery for a distal radius fracture was treated with UDPHAp implantation [11]. The UDPHAp implant can be easily shaped to fit the defect site and becomes fused with the surrounding bone within approximately three months. CT evaluation 6 months after surgery showed that the UDPHAp implant was uniformly composed of cortical bone adjacent to trabecular bone, and the articular surface of the distal radius was preserved. A bone sample of the implanted UDPHAp was obtained during plate removal surgery. The histologic evaluation of the implant specimen revealed the presence of ossified bone stained green with Villanueva–Goldner stain. The patient had a favorable postoperative clinical course with no complications or impairment of daily activities. The unidirectional interconnected porous structure of UDPHAp increased the compressive strength of the material and allowed osteogenesis within the implant. Four patients with intra-articular calcaneal fractures underwent open reduction of the displaced fragments with block-type and granule-type UDPHAp implantation to preserve the articular surface [12]. The articular angle improved within the normal range after open reduction. The UDPHAp implants were incorporated into the surrounding bone, and trabecular bone was observed on CT 12 weeks after surgery. There was no dislocation or breakage of the UDPHAp implants during evaluation, indicating their stability and effectiveness in maintaining the articular surface. All patients had good postoperative function with no implant-related complications or loss of fracture correction. UDPHAp implantation has shown promising results in the repair of intra-articular fractures. These clinical results suggest that UDPHAp may be a useful bone graft substitute for filling bone defects during the open reduction of intra-articular fractures.

## 4. Discussion

It is common to fill the cavity left after intralesional resection to restore mechanical strength. Although bone defects after tumor resection may not routinely require bone filling [13,14], histologic evaluation showed that HAp implantation into defects could improve bone repair processes compared with blood clots alone as a control [15]. In addition, the implantation of bone graft substitutes can improve cortical bone thickness due to benign bone tumors [5]. Therefore, we believe that filling bone defects with HAp may provide early structural and biological benefits in the treatment of pathological fractures or cortical thinning caused by benign bone tumors. UDPHAp has an interconnected porous structure consisting of unidirectional oval pores in the horizontal direction that completely penetrate the material. The implantation of UDPHAp resulted in good bone formation in defects with few complications after the intralesional resection of benign bone tumors. In addition, cortical thickness generally increases after tumor resection and UDPHAp implantation. Although one patient with SBC fell down the stairs and refractured the bone 9 months after implant removal, we believe this was an accident and not related to the implanted UDPHAp. The patient underwent another internal fixation and was able to perform normal daily activities with excellent radiographic evaluation at the last follow-up. 

Interconnected porous calcium HAp (IP-CHA) was also developed as a second-generation HAp with excellent interpore connections, which allows superior osteoconduction by allowing cells and tissues to invade deep into the pores compared to conventional HAp, which has few interconnected pores [16]. The initial compression strength of IP-CHA was approximately 10 MPa. Previously, good early bone formation was observed after benign bone tumor resection with IP-CHA implantation [3]. UDPHAp implantation achieved more advanced radiographic stages at the final follow-up and shorter time to advanced stages, according to the radiographic evaluation of IP-CHA implantation [6]. In addition, a previous study confirmed that UDPHAp could facilitate early bone formation because it showed more cells, rhBMP-2, and vascularization in the pores than IP-CHA [17]. These findings suggest that UDPHAp can stimulate early and reliable bone formation and is a useful bone graft substitute for the treatment of benign bone tumors with a low complication rate.

Fibular grafting is one of the most common orthopedic procedures used to reconstruct bone defects. In particular, because large bone defects after bone tumor resection often require bone graft material of good strength and quality, a fibular graft is a good candidate. Although the reconstruction of the fibular defect after harvesting may not be necessary, the loss of the fibula after harvesting sometimes results in significant donor site morbidity [18,19]. Few studies have evaluated the potential of HAp as a bone graft substitute at fibular donor sites. To the best of our knowledge, only one comparative study (in Japanese) has shown that patients with conventional (first-generation) HAp implantation at the donor site had fewer complications related to fibular harvesting than those without HAp [20]. Currently, we believe that the regeneration of the fibula in the defect after graft harvesting is critical to minimize morbidities after fibular graft harvesting, especially in pediatric patients. However, 25% of patients treated with HAp spacers showed spacer breakage and/or wire stabilization of the spacer to the remaining fibula [20]. This finding suggests that bone ingrowth into the pores of the conventional HAp is insufficient for new bone formation. The implantation of UDPHAp in a segmental defect of the fibula results in rapid bone formation around the material. Good regeneration and continuity of the fibula was observed at the final follow-up. There are several possible reasons for these results. First, as discussed above, unidirectional pores in the horizontal direction with some interpore connections may facilitate cell and blood migration and invasion into the material, which is beneficial for new bone formation and implant regeneration. Second, the periosteum was preserved when harvesting the fibula, as an intact periosteum may be an important contributor to good bone formation [21]. Third, this was a pediatric patient who may have a good ability to facilitate osteogenesis. We believe that UDPHAp is an excellent bone substitute material for fibular regeneration that can effectively minimize morbidity after fibular graft harvesting, especially in pediatric patients.

Because of the unique properties of UDPHAp, histologic analyses showed good bone regeneration and remodeling within UDPHAp materials when implanted in animal bone. A cortical bone defect was created in rabbits, and UDPHAp was implanted into the cavity. New bone and capillaries were observed within the UDPHAp implants at 2 weeks, and new bone formation was identified in 41.6% of the porous area at 12 weeks [22]. These findings indicated the early stages of osteogenesis and successful bone regeneration inside UDPHAp materials. Osteon-like structures, characterized by the presence of lacunae, canaliculi, and capillaries, were observed within the implanted UDPHAp material at 6 weeks [23]. The rapid formation of unidirectional capillaries and the osteocyte lacunae–canalicular system may promote continuous bone remodeling. Moreover, long-term histological analysis confirmed that bone formation within the UDPHAp continued to be remodeled for up to two years [24]. A study on long-term UDPHAp implantation in canines also demonstrated that bone ingrowth and gradual resorption of the UDPHAp were observed 1 to 3 years after implantation, with replacement by trabecular bone [25]. HAp is widely considered a non-biodegradable material or a material that is not easily absorbed in the body. The findings from animal experiments and clinical cases demonstrated that the implanted UDPHAp can promote early bone formation and has the potential to be absorbed and replaced with the host bone. From a different point of view, the new bone extended from the border of cortical bone and UDPHAp implants into the center and intramedullary part of the implants [22]. The numbers of osteocytes and osteon-like structures were significantly higher in areas adjacent to the cortex of the host bone compared to areas next to the medullary cavity [23]. Long-term analysis demonstrated that new bone formation occurred at the contact sites with the cortical bone and then extended into the central and intramedullary portions of the material. These findings suggested that load transfer may affect the formation of osteocytes and osteon-like structures inside the UDPHAp implants and that mechanical stress from the surrounding cortical bone and angiogenesis may contribute to new bone formation [24]. Basic research on implantation in animals provides a comprehensive histologic analysis of long-term bone formation and remodeling within the UDPHAp at a cortical bone defect site, and these findings support the usefulness of UDPHAp as a bone substitute material in clinical settings.

Preclinical studies of UDPHAp implantation have also been conducted in animals. In canines, a wedge-shaped UDPHAp was implanted in the gap following a tibial wedge osteotomy, and radiographic evaluation showed complete consolidation and bony fusion at the osteotomy site and UDPHAp at 12 weeks, indicating the formation of a strong bond [26]. Histologic evaluation showed new bone formation and direct attachment at 12 weeks, indicating successful bone ingrowth and integration with the implant. These findings support the good radiographic and clinical results of open-wedge high tibial osteotomies treated with UDPHAp [8]. UDPHAp has a high compressive strength parallel to the unidirectional pores of 13.1 MPa, whereas other HAp products have a compressive strength of 2–10 MPa. Since good clinical results have been reported with this initial strength, postoperative weight bearing may be allowed earlier with UDPHAp. In goats, UDPHAp spacers were implanted between the split lumbar laminae, and although histologic analysis showed that the pore shapes in the UDPHAp spacers were altered, limited new bone formation was observed [27]. This result may be due to the low initial compressive strength of UDPHAp resulting from its higher porosity. This study concluded that maintaining the pore shape is necessary to enhance new bone formation in UDPHAp when used as a lamina spacer in spinal surgery. The findings on UDPHAp spacers in animal studies are consistent with the clinical findings on UDPHAp spacers for open-door laminoplasty [9]. The results of the animal studies also support the fact that UDPHAp spacers are not suitable for open-door laminoplasty.

To the best of our knowledge, there is only one histologic study of a human bone specimen implanted with UDPHAp, which was conducted one year after implantation in a case with a radial fracture. Histologic examination revealed ossified bone in the unidirectional pores of the implanted UDPHAp; however, there was no evidence of active bone metabolism [11]. This finding is not consistent with the previous results of animal studies in which active osteogenesis was observed in the early stage after implantation. This could be due to the completion of bone repair at the time of sampling and/or reduced bone metabolism associated with older age. Further histologic studies may provide additional evidence in human specimens. However, human samples for UDPHAp implantation are clinically difficult to obtain, mainly for ethical reasons. As shown by clinical radiographic evaluation, UDPHAp can stimulate early new bone formation and good remodeling within implants and is a useful bone graft substitute for orthopedic surgery.

Beta-tricalcium phosphate is also one of the most commonly used synthetic bone graft substitutes and is osteoconductive and resorbable [28]. Unidirectional porous beta-tricalcium phosphate (UDPTCP) was developed based on the same concept as the unidirectional porous structure of UDPHA, and initial compression strengths of 8 and 1.5 MPa are applied in the directions parallel and perpendicular to the pores, respectively. Several experiments have been performed on UDPTCP implantation in animals [2]. Histologic evaluation revealed new bone formation throughout the interior of the material six weeks after implantation. Host bone replacement was noted in the cortical bone region, and the resorption and remodeling of the material progressed in the medullary cavity. Clinically, a retrospective study was performed to radiographically evaluate the healing process of bone defects filled with UDPTCP in 56 patients treated for the resection of benign bone lesions [29]. Radiographic evaluation showed that the resorption of UDPTCP and bone trabeculation through the defect were 83% and 94%, respectively, 12 months after implantation in patients with tumors in bones other than the phalanges and metacarpal/tarsal bones. No complications, including infection, postoperative fracture, and allergic reaction, were associated with the use of UDPTCP. This study demonstrated that UDPTCP is a useful bone graft material for filling defects after the resection of benign bone tumors. UDPTCP was implanted to fill a bone defect in the treatment of distal radial fractures, and radiographic evaluation showed that UDPTCP implantation with internal fixation for distal radial fractures resulted in the resorption of the implant in the early postoperative period and replacement by the host bone less than one year after surgery [30,31]. In addition, UDPTCP filling resulted in the restoration and preservation of anatomic position after the correction of distal radius fractures in elderly patients. The range of motion of the wrist improved to the normal range, and the clinical results at the final follow-up were excellent. Another report of a distal femoral fracture treated with UDPTCP implantation also confirmed a similar finding of progressive replacement of the implant with host bone over time [32]. UDPTCP was implanted to fill the gap in open-wedge high tibial osteotomy, and imaging evaluation showed early resorption of the implant and excellent replacement by the host bone [33]. The remodeling of UDPTCP implanted for open-wedge high tibial osteotomy was detected on CT earlier than that of TCP with a spherical interconnected porous structure. Similar to UDPHAp, UDPTCP has acceptable clinical performance as a bone graft substitute in orthopedic surgery. The unidirectional porous structure facilitates the invasion of cells and fluids necessary for osteogenesis into the implant. The defect after fibula harvesting for bone grafting was filled with columnar UDPTCP implants. Radiographs showed bony fusion between the host fibula and implanted UDPTCP 6 months after surgery [32]. The regeneration of natural bone with a tubular structure was observed 2 years after surgery. However, there is no full continuity of the regenerated fibula at the UDPTCP implantation site, in contrast to the finding of clear formation of cortex and marrow with full continuity of the fibula treated with UDPHAp implantation [7]. This may be due to the difference in the age of the patients rather than the implant materials. One clinical study showed that implantation into a bone defect using tricalcium phosphate mixed with HAp gave similar results to those obtained with allografts alone [34]. It may be an interesting treatment strategy to fill the bone defect with a mixture of UDPHAp and UDPTCP.

## 5. Conclusions

UDPHAp implants have been used in various orthopedic procedures. Radiographic results showed rapid bone formation and good remodeling after bone tumor resection and filling the bone defect with UDPHAp. Cortical thickness typically increases after tumor resection and UDPHAp implantation. In the case of segmental defects of the fibula, UDPHAp implantation resulted in rapid bone formation around the material, excellent regeneration of the fibula, and effective reconstruction of the trabecular bone. Results from various orthopedic surgeries indicated that UDPHAp showed excellent osteoconductivity and new bone formation early after implantation. Histological analysis in animal studies showed good bone regeneration and remodeling within the UDPHAp. These results suggest that UDPHAp is a promising bone graft substitute for the treatment of benign bone tumors and other orthopedic conditions with a low complication rate. 

## 6. Future Directions

UDPHAp implantation is associated with early bone formation and low complication rates. The use of clinically applicable heating materials with heat stimuli using UDPHAp has been shown to significantly enhance osteogenesis in animal models, suggesting that it is a promising treatment option for diseases associated with bone defects [35]. Additional treatment options, such as bioactive molecules, should be explored to facilitate new bone formation in implants. Further clinical studies using UDPHAp in combination with bioactive molecules are warranted to determine their potential to improve clinical outcomes.

Biphasic calcium phosphates, consisting of HAp and beta-tricalcium phosphate, are widely used as synthetic implants for bone graft materials in orthopedic surgery and have been shown to be safe, effective, and biocompatible scaffolds for new bone formation [36]. A potential approach to enhance bone formation could be the creation of a novel bone graft substitute composed of HAp and beta-tricalcium phosphate that incorporates the structural feature of unidirectional oval pores oriented in a horizontal direction, which could be referred to as unidirectional porous biphasic calcium phosphates.

## Figures and Tables

**Figure 1 biomimetics-09-00294-f001:**
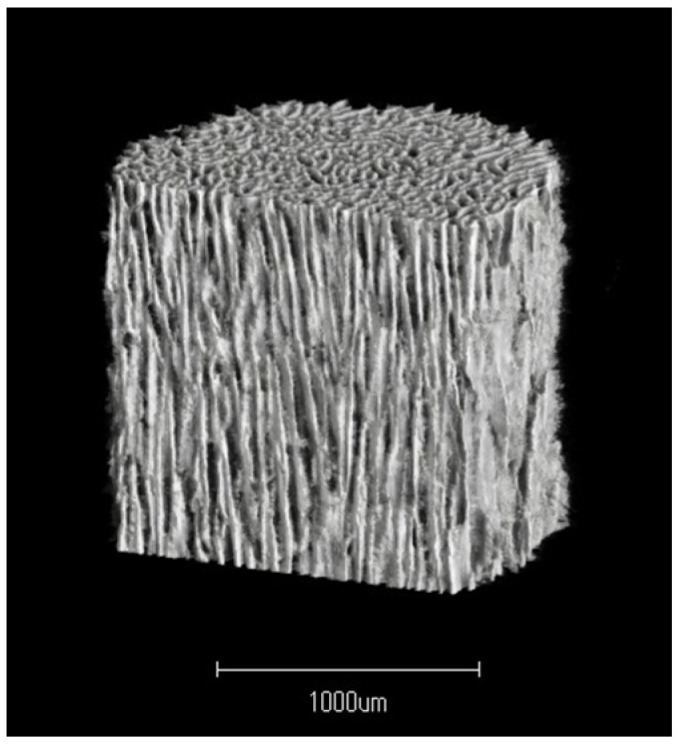
Three-dimensional micro-CT image of UDPHAp (provided by Kuraray Co., Ltd., Tokyo, Japan), showing the unidirectional pores in the vertical direction and some interconnection in the horizontal direction.

**Figure 2 biomimetics-09-00294-f002:**
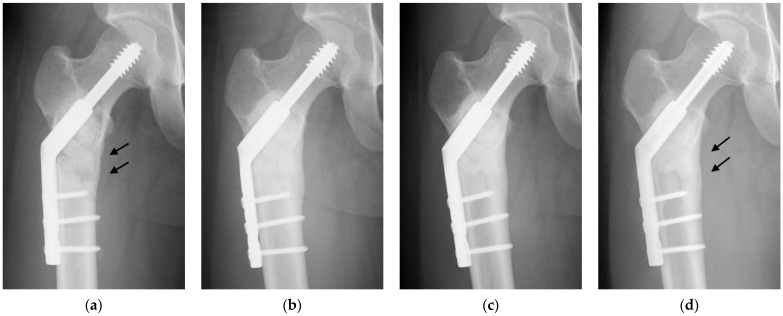
Radiographs of a 23-year-old simple bone cyst of the right proximal femur treated with UDPHAp [6]. (**a**) The UDPHAp implantation and internal fixation were shown on a radiograph 2 weeks postoperatively. A lytic change with cortical thinning of the medial bone cortex (arrows) was noted. Full weight bearing was allowed one day after surgery, and the patient returned to normal daily activities 2 months postoperatively; (**b**) moderate bone formation in UDPHAp was confirmed 7 months postoperatively; (**c**) the generation of medial cortical thinning was seen 13 months postoperatively; (**d**) UDPHAp resorption and bone marrow remodeling were observed 2 years and 6 months postoperatively. The decrease in medial cortical thickness was completely reversed (arrows).

**Figure 3 biomimetics-09-00294-f003:**
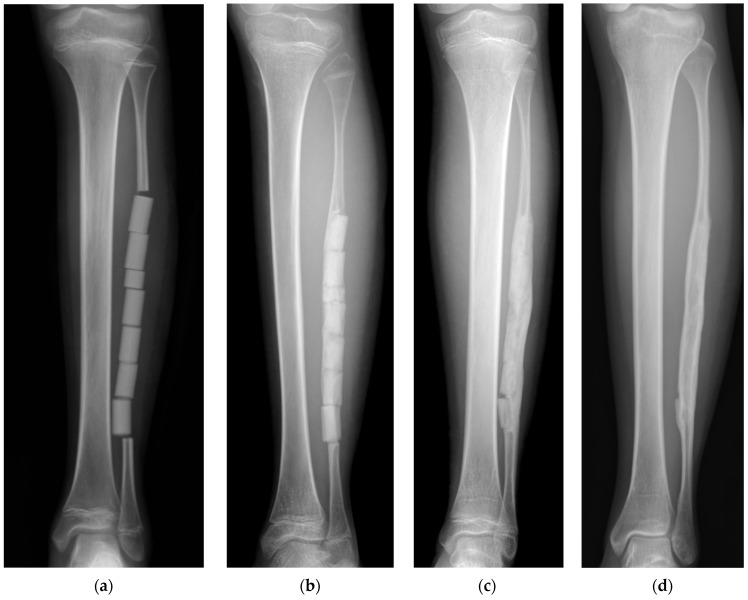
Radiographs of UDPHAp implantation in a segmental bone defect of the fibula after harvesting [7]. (**a**) Column-shaped UDPHAp implants were visible one week after surgery. Full weight bearing on the left leg was not allowed within 6 weeks after surgery to stabilize the pelvic reconstruction. The patient returned to normal daily activities 6 months postoperatively due to postoperative chemotherapy; (**b**) the resorption of the implanted UDPHAp was detected, and partial remodeling of the marrow space was seen 11 months after surgery; (**c**) new bone formation and resorption of implanted UDPHAp had progressed 2.5 years after surgery; (**d**) the complete resorption of implanted UDPHAp and clear formation of cortex and marrow were observed 12 years after surgery.

**Table 1 biomimetics-09-00294-t001:** Clinical factors influencing radiographic assessment at the final follow-up [6].

Characteristics	n	Final Radiographic Assessment		*p* Value
		Stage 3	Stage 4	
Total	44	24	20	
Male	30	19	11	0.09
Female	14	5	9	
Age ≤ 15 years	17	6	11	0.04
>15 years	27	18	9	
Long tubular bone	33	17	16	0.48
Non-long tubular bone	11	7	4	
Site in long tubular bone				
Including diaphysis	16	11	5	0.06
Metaphysis or epiphysis only	17	6	11	
Pathological fracture (+)	9	3	6	0.15
(−)	35	21	14	
UDPHAp volume ≤ 5 g	31	17	14	0.95
>5 g	13	7	6	
Follow-up period (months)				
≤36	23	15	8	0.14
>36	21	9	12	

## Data Availability

No new data were created or analyzed in this study. Data sharing is not applicable to this article.

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
