# Peer review of "Clinical Application of Unidirectional Porous Hydroxyapatite to Bone Tumor Surgery and Other Orthopedic Surgery"

_biomimetics, 2024, doi:10.3390/biomimetics9050294_

Round 1

Reviewer 1 Report

Comments and Suggestions for Authors

The study submitted by Kunisada et al. aims to assess the radiographic and clinical outcomes of patients who received UDPHAp implantation following surgical resection of benign bone tumors. Additionally, it reviews literature documenting clinical case results of UDPHAp implantation in various other orthopedic procedures. Overall, the paper is well-written and informative. However, there are some concerns to be addressed before considering publication.

Abstract

The abstract needs to highlight the background of the study

Introduction

Figure 1 was previously published in other papers. No reference regarding source or copyright appears in the legend.

Clinical Application to Bone Tumor Surgery

Some statistical analysis data should be further produced such as the inclusion criteria, appropriate measures taken to remove bias and the inter-observer agreement assessment. Also, limitations of the methods should be presented. Furthermore, the sensitivity and specificity of radiographs is inferior to CT and this should be addressed.

Could the authors provide additional details regarding the UDPHAp utilized, including its brand, any challenges encountered during manipulation, and the accessibility of the product? Are there any postoperative follow-up protocols being utilized?

Could the results be presented in table format?

In the outlined examples, the time to full weight bearing and full recovery is not highlighted.

Figure 2 contains images previously published in another paper. No copyright or reference is provided in the legend. 

Clinical Application to Other Orthopedic Surgery

Similar observations to the previous section in regard to values for both UDPHAp spacers and autologous bone spacers, inter-observer agreement and limitations. Can we also have a table in this section with the descriptive presentation of the results?

Line 157 – JOA score is a abbreviation. Please provide the full name for this score – Japanese Orthopaedic Association score, as this is not mentioned previously.

Discussions

At some point, the discussions focus towards unidirectional porous beta-tricalcium phosphate, mentioning several retrospective studies regarding the use of UDPTCP. However interesting, the purpose of this discussion is not clear, as there is no comparison provided to UDPHAp in terms of the role in bone recovery. Also, the authors should consider expanding the discussion using the following suggested references: DOI: 10.1016/j.actbio.2020.06.022, DOI: 10.3892/etm.2020.9345, DOI: 10.1002/jbm.b.34049

Last two paragraphs are the same (line 313-384).

Comments on the Quality of English Language

Language is fine, it requires only minor editing.

Reviewer 2 Report

Comments and Suggestions for Authors

The authors review the application of UDPHAp primarily in the surgical treatment of bone tumors and report the authors' clinical results. In addition, they review the results of other surgical procedures such as spine surgery and high tibial osteotomy, and discuss the usefulness of UDPHAp in orthopedic surgery. The review is well organized with a wide range of findings and discussion of UDPHAp. In the Discussion section, however, the authors repeat much of the same information as in the previous sections regarding the properties of UDPHAp. In addition to the description of UDPHAp, the characteristics and clinical applications of IP-CHA and UDPTCP are also described, making the section longer and less coherent. The following is a list of items that should be corrected.

1

The items to be discussed can be reorganized and similar descriptions can be omitted. In particular, Lines 198 - 211 are similar to the contents already described in other sections, and it is desirable to summarize the contents since the Discussion section is long.

2

If the characteristics and clinical applications of IP-CHA and UDPTCP and their therapeutic outcomes are to be discussed, it would be better to mention them in other sections. It would be easier to understand if the structural characteristics, mechanical strength, and clinical applications of UDPHAp, IP-CHA, and UDPTCP are summarized in a table.

3

Please also introduce the cause of the difference in fibula regeneration between those implanted with UDPHAp and those implanted with UDPTCP regarding Lines 343-345 and describe the difference in these characteristics.

Comments on the Quality of English Language

No specific comments.

Reviewer 3 Report

Comments and Suggestions for Authors

In my humble opinion, this well-written manuscript might be published almost as is. Just add a caption to Fig. 1.

Reviewer 4 Report

Comments and Suggestions for Authors

A review of the clinical applications of unidirectional porous hydroxyapatite (UDPHAp) is of significant interest in various surgical fields for the treatment and repair of bone defects and fractures. Unfortunately, the publication requires revision for a number of reasons.

Firstly, the commercial brand UDPHAp is not represented.

Secondly, the given Figures are partially duplicated by the Figures in article [6] without references.

Third, Figure 1 is not captioned.

Fourthly, there is information that is duplicated in the text of the article. For example, the information in lines 313–348 and 349–384.

Round 2

Reviewer 1 Report

Comments and Suggestions for Authors

The authors have adequately addressed the previously reported issues.

Some minor English editing is still required.

Also, perhaps it would be possible to align all image fragments from Figure 2, because in the current form in appears disorganized.

Best regards,

Comments on the Quality of English Language

Some minor English editing is still required.

Author Response

Thank you for taking the time to review our manuscript. We are grateful for your positive feedback. We have already had our manuscript reviewed by a local English editing service. We would like the editorial team to handle the minor English editing you mentioned. Additionally, the necessary modifications to the images in Figure 2 will also be taken care of by the editorial team.

Reviewer 2 Report

Comments and Suggestions for Authors

No specific comments.

Author Response

Thank you for taking the time to review our manuscript. We greatly appreciate your favorable comments.

Reviewer 4 Report

Comments and Suggestions for Authors

The Manuscript  may be published after minor revision and editorial revision of the quality of English.

Author Response

Thank you for taking the time to review our manuscript. We greatly appreciate your favorable comments.

We have already had our manuscript reviewed by a local English editing service. We would like the editorial team to handle the minor revision and editorial revision of the quality of English you mentioned.